# Sildenafil during the 2nd and 3rd Trimester of Pregnancy: Trials and Tribulations

**DOI:** 10.3390/ijerph191811207

**Published:** 2022-09-06

**Authors:** Felix Rafael De Bie, David Basurto, Sailesh Kumar, Jan Deprest, Francesca Maria Russo

**Affiliations:** 1Department of Development and Regeneration, KU Leuven, 3000 Leuven, Belgium; 2Mater Research Institute and School of Medicine, University of Queensland, Brisbane, QLD 4343, Australia; 3Department of Obstetrics and Gynecology, UZ Leuven, 3000 Leuven, Belgium

**Keywords:** sildenafil citrate, 2nd- and 3rd-trimester pregnancy, fetal therapy, pulmonary arterial hypertension, preeclampsia, preterm labor, fetal distress, oligohydramnios, fetal growth restriction, congenital diaphragmatic hernia

## Abstract

Sildenafil, a phosphodiesterase 5 inhibitor with a vasodilatory and anti-remodeling effect, has been investigated concerning various conditions during pregnancy. Per indication, we herein review the rationale and the most relevant experimental and clinical studies, including systematic reviews and meta-analyses, when available. Indications for using sildenafil during the second and third trimester of pregnancy include maternal pulmonary hypertension, preeclampsia, preterm labor, fetal growth restriction, oligohydramnios, fetal distress, and congenital diaphragmatic hernia. For most indications, the rationale for administering prenatal sildenafil is based on limited, equivocal data from in vitro studies and rodent disease models. Clinical studies report mild maternal side effects and suggest good fetal tolerance and safety depending on the underlying pathology.

## 1. Sildenafil

Sildenafil or sildenafil citrate selectively inhibits phosphodiesterase 5 (PDE5), an enzyme found in high concentration in the endothelium and smooth muscle cells of the corpora cavernosa, pulmonary arterial smooth muscle cells, and platelets [1,2,3,4,5]. PDE5 degrades cyclic guanosine monophosphate (cGMP), a crucial mediator in the nitric oxide (NO) pathway (Figure 1).

Endothelium-derived NO stimulates the formation of cGMP in vascular smooth muscle cells, which has a double effect: (1) modulation of ion channel activity causing vasodilation, and (2) formation of cGMP-dependent protein kinase (PKG) inhibiting smooth muscle cell proliferation [6]. Inhibition of PDE5 increases intracellular cGMP levels, producing vasodilatory and anti-remodeling effects.

Pfizer initially investigated sildenafil in the mid-1980s as a potential alternative to nitrates for treating angina pectoris. In phase I clinical trials, healthy volunteers reported dose-congruent side effects, including headaches, flushing, muscle cramps, indigestion, transient disturbances of color vision perception (due to weak inhibition of PDE6 in retinal photoreceptors), and penile erections [7]. The latter side effect eventually granted the compound its fame since sildenafil became an attractive alternative to the invasive, intracavernosal injections of vasodilators, and it was proven effective for almost all types of erectile dysfunction (diabetes, cardiovascular disease, multiple sclerosis, spinal cord injury). Sildenafil obtained market approval from the Food and Drug Administration (FDA) and European Medicines Agency (EMA) in 1998. Since then, sildenafil was also approved by the FDA and EMA for adult and pediatric pulmonary arterial hypertension (PAH) and has been investigated for a wide range of indications, such as Raynaud’s disease, heart failure, cardiac and cerebral ischemia, as well as infarction [7,8].

The rationale of sildenafil to treat these conditions can be understood in terms of sildenafil’s capacity to reverse endothelial dysfunction, prevent vascular smooth muscle remodeling, and selectively improve regional blood flow in areas of greatest need using vascular smooth muscle relaxation [7].

These effects could be particularly interesting for maternal or fetal pathology during pregnancy. Sildenafil only modestly reduces blood pressure, granting it a particularly desirable profile for pregnancy where systemic hypotension is deleterious for the fetus [7,9]. Moreover, the hepatic CYP3A4 and CYP2C19 isoenzymes involved in sildenafil’s metabolization were present and active prenatally [10,11]. Tadalafil, vardenafil, mirodenafil, avanafil, and udenafil are other PDE5 inhibitors, however only tadalafil has also been trialed during pregnancy for fetal growth restriction [12,13]. Compared to sildenafil, clinical experience with tadalafil is very limited, and immaturity of the glucuronidation pathway used for its metabolization forms a contraindication for its use during pregnancy [14].

While sildenafil has been increasingly proposed and investigated for the treatment of female infertility [15,16,17] and the prevention of recurrent miscarriages [15,18,19,20], this review only discusses the use of sildenafil in the second and third trimesters of pregnancy.

For most indications, sildenafil was considered because of its local vasodilatory effect on addressing compromised pulmonary (maternal pulmonary hypertension) or uteroplacental perfusion (preeclampsia, fetal growth restriction, and fetal distress). Its antiremodeling effect was hypothesized to reduce vascular pulmonary (congenital diaphragmatic hernia) or uteroplacental (preeclampsia) maldevelopment. Because of its smooth muscle relaxant effect, sildenafil was also tried as a tocolytic for preterm labor.

Hereunder we review the pathophysiologic rationale for using sildenafil per indication, as well as the most relevant experimental and clinical studies.

## 2. Sildenafil Use during Pregnancy

Epidemiological studies conducted in the United States and Scandinavian countries linked prescription fills for PDE5-inhibitors and pregnancy data using insurance data and national registries, respectively, mapping the use of sildenafil during pregnancy [21,22]. Both studies demonstrated that only a minority of patients using PDE5-inhibitors during pregnancy had a diagnosis of PAH (4–16%), and therefore most sildenafil use during pregnancy was off-label.

## 3. Maternal Pulmonary Hypertension

### 3.1. Rationale

PAH is a progressive disease preferentially affecting women of childbearing age. It is characterized by elevated pulmonary vascular resistance, which, when aggravated by the physiologic changes in pregnancy and during parturition, can precipitate right ventricular failure leading to maternal and fetal complications and even mortality [23].

### 3.2. Clinical Evidence

While pregnant patients are systematically excluded from PAH clinical trials, some case series report on the chronic use of sildenafil during pregnancy, suggesting efficacy and safety without apparent maternal or fetal side effects [24,25,26,27,28].

## 4. Preeclampsia

### 4.1. Rationale

Preeclampsia is a pregnancy complication typically occurring after 20 weeks of gestation due to abnormal placentation. Defective invasion of the uterine arteries by the extra-villous placental cytotrophoblasts causes placental secretion of antiangiogenic factors causing endothelial dysfunction [29]. In addition to altered remodeling of spiral arteries, increased sensitization of maternal vessels to the vasoconstrictive effects of angiotensin II’s (AT-II) and decreased activity of the nitric oxide (NO) vasodilatory pathway have been demonstrated in preeclampsia [29,30,31]. The clinical syndrome of preeclampsia is characterized by new-onset maternal hypertension, proteinuria, other signs of maternal organ dysfunction (hepatic, renal, neurological, and hematologic), and placental hypoperfusion compromising fetal growth [9]. Currently, the only cure is the termination of pregnancy to deliver the placenta, often resulting in preterm birth [32]. An antenatal treatment mitigating the vascular changes can potentially prolong pregnancy and is, therefore, more than welcome. Since teratogenicity prevents the use of angiotensin-converting enzyme inhibitors to decrease the production of AT-II, sildenafil has been proposed as a good candidate to restore endothelial homeostasis by increasing NO-mediated vasodilation and its anti-remodeling effects [33,34].

### 4.2. Experimental Evidence

Several studies in pregnant mice and rat models of preeclampsia have demonstrated the beneficial effects of sildenafil on maternal blood pressure, micro-albuminuria, glomerular injury, fetal weight, fetal survival, and even offspring learning capacities [9,35,36]. A recent ex-vivo study of human preeclamptic placentas by Hitzerd et al. demonstrated increased transplacental transfer of sildenafil and no potentiation of NO-mediated vasodilation (chorionic plate arteries) in preeclamptic placentas. Although this suggests that sildenafil may not exert a direct vasodilatory effect, it remains to be examined whether it may offer benefit through its antiremodeling effect [32].

### 4.3. Clinical Evidence

Two randomized controlled trials (RCT) investigated whether sildenafil could prolong pregnancy in case of preeclampsia [37,38] (Table 1). One RCT demonstrated a reduction in average maternal blood pressure but no effect on gestational age at birth [37]. In the other RCT, in which more patients were included (100 vs. 39), sildenafil was administered earlier in gestation (29^+1^ vs. 31^+4^ weeks) and at a higher dose (50 mg vs. 20 mg three times daily), and demonstrated prolongation of pregnancy by on average four days [38]. Both studies reported mildly decreased maternal blood pressures and no sildenafil-associated maternal, fetal or neonatal safety issues.

## 5. Preterm Labor

### 5.1. Rationale

Although preterm labor has multiple known risk factors (short cervix, multiple pregnancies, infection, smoking, age > 18 y or >35 y, etc.), its exact pathomechanism remains unclear [39]. The main concern is that preterm labor leads to the birth of a developmentally immature infant with consequent severe morbidity. Few therapeutic tools exist other than tocolytics to suppress uterine contractions and allow for corticosteroid-induced acceleration of lung maturation, neuroprotection with magnesium sulfate, and transfer to a tertiary care facility if needed [40,41]. Sildenafil was hypothesized to induce uterine quiescence in patients with threatened preterm delivery through NO-induced smooth muscle relaxation [42,43].

### 5.2. Experimental Evidence

Sildenafil was shown to reduce myometrial contractility by measuring intrauterine pressures in a rat model in preterm labor [42]. The effect was confirmed in-vitro using human myometrial samples [43]. However, the very high concentrations of sildenafil required for direct myometrial relaxation were expected to limit clinical translation because of its hypotensive and other cardiovascular side effects [44]. Nonetheless, lower therapeutic concentrations of sildenafil sensitized in-vitro human myometrium to the tocolytic calcium-antagonist nifedipine [45].

### 5.3. Clinical Evidence

One clinical trial investigating sildenafil as an add-on therapy to nifedipine, a calcium-antagonist, for threatened preterm labor was retracted due to concerns about the validity of the data [46]. An Iranian study conducted in 2020 suggested a beneficial effect of adding sildenafil to nifedipine therapy for threatened preterm labor reporting prolonged latency (16 days instead of 10 days), as well as the reduced prevalence of neonatal respiratory distress syndrome (7.6% vs. 25.8%) and higher birth weights (2154 g vs. 1609 g) [47] (Table 2).

## 6. Fetal Growth Restriction

### 6.1. Rationale

Severe early-onset fetal growth restriction (FGR) complicates 0.4% of pregnancies and carries a high risk of perinatal morbidity and mortality, primarily due to premature delivery [48,49]. Placental insufficiency can lead to suboptimal delivery of oxygen and nutrients to the fetus, stunting growth and development. Current management consists of intensive fetal and uteroplacental circulation monitoring, with timely delivery in case of threatening compromise [50].

NO-synthase deficiency and decreased NO bioavailability lead to impaired NO-mediated vasodilation in FGR fetuses [51]. Sildenafil in FGR is hypothesized to enhance the NO-mediated effects, facilitating vascular adaptations essential to pregnancy, i.e., reducing maternal peripheral vascular resistance to generate a low-resistance/high-caliber uteroplacental unit to ensure blood flow to the fetus [52,53].

### 6.2. Experimental Evidence

Ex vivo wire-myography analysis of myometrial arteries originating from women with FGR was shown to be more vasoconstricted with less endothelium-dependent vasodilation than normal controls. When exposed to sildenafil, FGR-myometrial arteries showed reduced vasoconstriction and improved relaxation [54].

In mouse models of FGR, multiple studies demonstrated improved uteroplacental blood flow and increased average fetal weights at birth in sildenafil-exposed animals [55,56,57,58]. However, more recent studies in FGR-mice investigating the long-term cardio-metabolic effects of prenatal sildenafil treatment could not confirm increased fetal growth and even demonstrated worse metabolic (disrupted glucose homeostasis) and cardiovascular outcomes (hypertension) in adult sildenafil-exposed offspring [59,60].

In rat models of FGR, sildenafil did not improve birth weights in one study [61], and mildly improved cardiovascular but not renal function in another study [62].

In a rabbit diet-based FGR model, prenatal therapy with sildenafil improved placental vascularity and fetal growth; however, it did not prevent FGR-related increased umbilical artery resistance and induced relative cerebral vasoconstriction [63].

In one fetal lamb study, where FGR was induced by single umbilical artery ligation, sildenafil reduced uteroplacental perfusion and induced fetal hypoxia, hypotension, and tachycardia [64]. In more recent lamb studies, again using single umbilical artery ligation, sildenafil exacerbated growth restriction and generated endothelial dysfunction in cerebral and femoral arteries [65,66]. The same group also demonstrated that sildenafil compromised the ‘brain-sparing’ cardiovascular adaptations in fetal lambs with FGR [66]. However, in another longer-term fetal lamb study, using uterine artery embolization to induce FGR, sildenafil was associated with increased placental and fetal weights following exposure to sildenafil for 21 days [67].

### 6.3. Clinical Evidence

In 2011, a first small non-randomized study from Canada demonstrated improved growth velocity in 10 women with severe, early-onset FGR and a tendency towards improved perinatal, intact survival without maternal adverse effects [68] (Table 3). Soon after, an RCT conducted in Iran reported an increased pulsatility index (PI) of the middle cerebral artery (MCA), suggesting less brain-sparing, as well as a decreased PI of the umbilical artery (UA), suggesting decreased umbilical vascular resistance, two hours after sildenafil administration [69]. Although similar Doppler findings were also reported in two other studies, these must be viewed with caution as both have had significant data integrity and trial governance concerns raised and are currently under investigation [70,71,72].

Together with an encouraging meta-analysis on the experimental use of sildenafil in FGR [35], and the beneficial effects in these clinical studies were the impetus for the creation of a formal international *Sildenafil Therapy in Dismal Prognosis Early-Onset Fetal Growth Restriction* (STRIDER) study consortium, with multiple study arms referred to by geographical location (United Kingdom, Ireland, the Netherlands, New Zealand, Australia, and Canada), each assessing different outcomes [77,78].

The British arm was the first to be completed. It did not demonstrate the benefit of low-dose sildenafil (25 mg 3 times daily) in terms of survival, perinatal morbidity, pregnancy duration, or birthweight [73]. Analysis of maternal hemodynamic effects of sildenafil revealed modest, short-term increased heart rate, reduced blood pressure, and reduced arterial stiffness [79]. Post-hoc review of fetal and neonatal deaths revealed no obvious differences in clinically relevant pathological and histopathological findings between sildenafil of placebo arms of the study [80].

The New Zealand and Australia arms followed, and also demonstrated no benefit regarding growth velocity and morbidity-free survival [74]. Analysis of a small subset of patients did not find an association between prenatal sildenafil therapy and neonatal cardiac dysfunction or pulmonary hypertension [81].

The Dutch trial was stopped early (216/360 women recruited), after an interim-analysis raised concerns about higher than expected neonatal mortality (24.7% (21/85) vs. 14.1% (11/78), *p* = 0.100) and high rates of persistent pulmonary hypertension of the newborn (PPHN) (18.8% (16/85) vs. 5.2% (4/78), *p* = 0.008) [75]. In addition, there were also concerns about futility, with therapeutic benefit considered unlikely if the study was continued [75].

In light of these concerns, the consortium published a physician alert to advise against the prescription of sildenafil for treating FGR, which led, among other things, to the early termination of the Canadian trial [82]. The findings and the halting of the STRIDER trials were widely covered by public media. They sparked an academic polemic about the potential pathophysiologic etiology of the findings [83,84,85], the robustness of supporting preclinical data [86], and the fallout on other indications for which antenatal sildenafil was considered [87,88]. The trial’s authors recently reported in more detail on the patients with pulmonary hypertension (PH) and deceased neonates, describing a higher incidence of early PH in sildenafil-exposed cases, yet without difference in late-onset PH or PH-related mortality [89]. Finally, the group also reported on the use of near-infrared spectroscopy in a very small subset of newborn patients, which revealed that sildenafil-treated patients had lower renal but not cerebral oxygenation during the first 72 h of life [90].

While awaiting a formal meta-analysis by the STRIDER consortium, another clinical study demonstrated increased MCA PI and decreased UA PI in FGR pregnancies treated with sildenafil [76], findings confirmed in a recent meta-analysis [91].

## 7. Oligohydramnios

### 7.1. Rationale

Oligohydramnios refers to the amniotic fluid volume that is less than the minimum expected for the gestational age. Whereas the exact mechanism of oligohydramnios remains unknown, the most probable etiology (maternal, fetal, placental, or idiopathic) depends on the oligohydramnios severity and gestational age at diagnosis. The amniotic fluid comprises a complex regulation mechanism that reflects maternal–fetal well-being and is key for fetal development. Pregnancies complicated by oligohydramnios are at risk for adverse perinatal outcomes [92,93]. During the second half of gestation, the main routes of amniotic fluid production include fetal urine and pulmonary secretions [93].

### 7.2. Clinical Evidence

Without preceding experimental evidence, an RCT was conducted with the hypothesis that sildenafil administration may result in increased fetal renal blood flow due to improved uteroplacental perfusion, hence improving urine production [94]. The study was retracted in April 2020 after an *expression of concern* was emitted in August 2019. The independent statistical reviewer concluded that there were patterns and correlations in the data set that were compatible with the invention of data [94,95]. To our knowledge, there is no other clinical or experimental data to support the use of sildenafil in the context of oligohydramnios.

## 8. Fetal Distress

### 8.1. Rationale

Most cases of fetal distress during labor can be attributed to inadequate placental perfusion or preexisting placental deterioration that impairs metabolic exchange with the fetus [96]. During contractions, the uterine blood flow can drop by more than half, which may result in intrapartum fetal compromise [97]. When fetal distress is evidenced, emergency operative delivery is the therapeutic option. Hypothetically, sildenafil could, through its vasodilatory action, improve uteroplacental perfusion and hence reduce the risk of emergency delivery for fetal distress.

### 8.2. Clinical Evidence

The randomized controlled RIDSTRESS trial (“*Reducing the risk of fetal distress with sildenafil study*”) investigated intrapartum sildenafil administration for the prevention of fetal distress [98]. The experimental evidence supporting the use of sildenafil in this context was based on animal studies in FGR models. The RIDSTRESS trial demonstrated a reduction of the risk of emergency operative birth by 51% and a reduction of the risk of meconium-stained liquor or pathologic fetal heart rate patterns by 43%, without differences in maternal or neonatal adverse events [99] (Table 4). Although very encouraging, the trial was not powered to show differences in neonatal outcomes, which can only be addressed in a larger phase III trial [100].

## 9. Congenital Diaphragmatic Hernia

### 9.1. Rationale

Congenital Diaphragmatic Hernia (CDH) is a rare congenital anomaly occurring in 1 of every 2000–4000 live births. Failure to completely form the diaphragm in the first trimester of pregnancy results in a diaphragmatic defect allowing abdominal organs to herniate into the chest and compress the developing lungs and heart [6]. This results in a trifecta of pulmonary hypoplasia, pulmonary hypertension, and cardiac hypoplasia, characterized by smaller and fewer alveoli with thicker walls and more interstitial tissue, a hypoplastic vascular bed with neo- and hyper-muscularized pulmonary arterioles, and predominantly left ventricle hypoplasia [101,102,103]. These rigid, hyper muscular pulmonary arterioles impede the extensive pulmonary vasodilation required for perinatal transition, which clinically translates into often lethal pulmonary hypertension [6]. The ability to diagnose CDH prenatally and its progressive nature form the rationale for modifying pathophysiology before birth [104]. Through its anti-remodeling effect, sildenafil is hypothesized to modulate vascular smooth muscle proliferation, reduce pulmonary arteriolar wall thickness, and reduce the pulmonary hypertensive phenotype of CDH [6].

### 9.2. Experimental

The above hypothesis was first tested by a Canadian group in a nitrofen rat model with congenital diaphragmatic hernia [105]. Maternal sildenafil administered during the embryonic period of lung development increased pulmonary vessel density, reduced vessel wall thickness, and right ventricular hypertrophy [105]. Over the years, these findings were confirmed by other groups [106,107,108] and were maintained when sildenafil exposure was delayed until a clinically more relevant lung development phase, i.e., the pseudoglandular phase [109,110]. An unexpected finding in these rat studies was that in control rats with non-hypoplastic lungs, sildenafil caused structural changes, i.e., decreased the pulmonary vascular volume and increased the pulmonary arterial smooth muscle wall thickness [105,106,107,108].

In a surgical rabbit model with CDH, sildenafil reduced the proportional wall thickness of peripheral lung vessels to the normal range, increased vascular branching, improved airway morphometry, and postnatal lung mechanics [111]. Combined with tracheal occlusion, sildenafil’s beneficial effect on airway development and lung function was augmented in pups with CDH [112]. Similarly to the findings in the rat studies, sildenafil reduced pulmonary vascular branching in non-hypoplastic littermates [111].

In a fetal lamb model with CDH, prenatal sildenafil exposure reduced pulmonary arterial pressure and increased pulmonary blood flow after birth, resulting in improved gas exchange [113]. However, the transplacental transfer of sildenafil to the fetal compartment in sheep was later shown to be very minimal [114]. This is tied to this animal model. In humans, this is likely to be different. In an ex-vivo dually perfused human model, sildenafil passed at a relatively high rate which was sufficient to reach clinically active fetal drug levels [115].

### 9.3. Clinical

These promising experimental results prompted a phase I/IIb clinical trial to investigate maternal and fetal safety and determine the pharmacokinetic profile of sildenafil in pregnancy [116]. Unfortunately, this study was suspended because of the safety concerns voiced by the Dutch STRIDER trial, despite the authors’ plea to not simply extrapolate the FGR findings to other clinical indications [87].

## 10. Safety and Tolerance of Sildenafil during Pregnancy

Whether sildenafil during pregnancy is safe was addressed in two systematic clinical reviews. Both studies agreed on the occurrence of mild maternal side effects, i.e., headaches, flushing, visual disturbances, dizziness, palpitations, arthralgia, dyspepsia, and epigastric pain, though not an increased risk compared to placebo [12,117]. Nasal bleeding was the only side-effect significantly more associated with sildenafil therapy (RR 10.53, 95% CI 1.36–81.3) [12]. Hypotension, a maternal side-effect of sildenafil that was anticipated, was not reported [117].

Obstetric outcomes depended on the underlying pathologic condition. Sildenafil reduced the risk of operative birth when used for intrapartum fetal distress and was not associated with new-onset pre-eclampsia, FGR, or antepartum hemorrhage [12].

Fetal and neonatal outcomes of prenatal sildenafil also depended on the underlying condition, as detailed above. Prenatal sildenafil was associated with an increased risk of persistent pulmonary hypertension only for FGR because of the findings in the Dutch STRIDER arm [12,75]. Meta-analysis could not find evidence for increased perinatal or neonatal mortality and morbidity (respiratory distress syndrome, necrotizing enterocolitis, intracranial hemorrhage, and retinopathy of the premature) associated with the use of PDE5-inhibitors during pregnancy [12].

## 11. Limitations

This review is limited in that it is not a systematic review in which the *Preferred Reporting Items for Systematic Reviews and Meta-Analyses (PRISMA)* guidelines were not strictly adhered to. Only the Medline database using PubMed was queried, and we also discussed non-randomized clinical trials.

## 12. Conclusions

Sildenafil has been used during pregnancy and investigated for a wide variety of indications. For most indications, the rationale for administering prenatal sildenafil is based on limited data from in vitro studies and low-complexity animal models. The lack of efficacy in some clinical studies further stresses the importance of completing the entire translational pathway sequentially. Independent of the indication, clinical studies report mild maternal side effects. However, fetal tolerance and safety outcomes were dependent on the underlying pathology.

## Figures and Tables

**Figure 1 ijerph-19-11207-f001:**
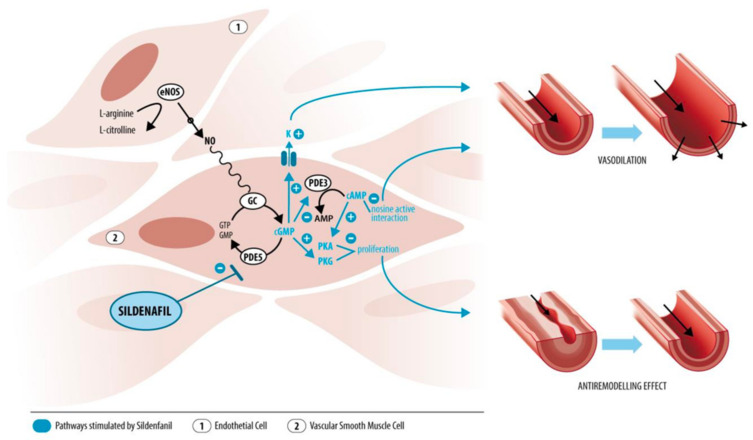
Nitric oxide pathway. Schematic representation of the mechanism of action of sildenafil on the pulmonary vasculature. eNOS, endothelial nitric oxide synthase; NO, nitric oxide; GC, guanylate cyclase; GTP, guanosine triphosphate; GMP, guanosine triphosphate; cGMP, cyclic GMP; PDE, phosphodiesterase; ATP, adenosine triphosphate; AMP, adenosine monophosphate; cAMP, cyclic AMP; PKG, cGMP-dependent protein kinase; PKA, cAMP-dependent protein kinase. Reprinted with permission from [6], 2022, Bentham Sciences. Drawing: Myrthe Boymans.

**Table 1 ijerph-19-11207-t001:** RCT’s assessing sildenafil for preeclampsia.

Author (Year) [Reference], Study Type—Country	Population [n = Sildenafil/Total]	Sildenafil Dosing	Gestational Age at Administration	Results	Study Limitations
Samangaya et al. (2009) [37], RCT—United Kingdom	Early-onset preeclampsia [n = 19/39]	20–80 mg, 3/day, PO	24–34 weeks	Sildenafil reduced maternal blood pressure but did not prolong pregnancy. No safety issues were observed regarding neonates.	Low patient number, low sildenafil dose, short administration duration (<15 days), and late advanced gestation at inclusion ~31^+4^ weeks.
Trapani et al. (2016) [38], RCT—Brazil	Early-onset preeclampsia [n = 50/100]	50 mg, 3/day, PO	24–33 weeks	Sildenafil increased pregnancy duration by four days, and reduced maternal mean arterial pressures and resistance in uterine and umbilical arteries. Pulsatility indices of the fetal middle cerebral artery were unaffected by sildenafil. No safety issues were observed regarding neonates.	Lack of power to compare neonatal outcomes, late gestation at inclusion ~29^+1^ weeks.

RCT = randomized controlled trial, PO = per os/oral.

**Table 2 ijerph-19-11207-t002:** RCT’s assessing sildenafil for preterm labor.

Author (Year) [Reference], Study Type—Country	Population [n = Sildenafil/Total]	Sildenafil Dosing	Gestational Age at Administration	Results	Study Limitations
Mohammadi et al. (2021) [47], RCT—Iran	Threatened preterm labor [n = 66/132]	10 mg nifedipine 3–4/day, PO + 25 mg sildenafil 3/day, PV	26–34 weeks	Nifedipine combined with sildenafil reduced delivery within seven days of admission, prolonged latency, and reduced prematurity compared to nifedipine alone.	Limited description of baseline group characteristics, missing sample size calculation, published in the group’s institutional journal (Tehran University of Medical Sciences), safety data not reported.

RCT = randomized controlled trial, PO = per os/oral, PV = per vaginam/vaginal.

**Table 3 ijerph-19-11207-t003:** Clinical studies assessing sildenafil for FGR.

Author (Year) [Reference], Study Type—Country	Population [n = Sildenafil/Total]	Sildenafil Dosing	Gestational Age at Administration	Results	Study Limitations
Von Dadelszen et al. (2011) [68], CS—Canada	Severe, early-onset FGR^1^ [n = 10/27]	25 mg, 3/day, PO, until birth	24–34 weeks	Sildenafil increased fetal abdominal circumference growth velocity. No maternal side effects were noted.	Small sample size, non-randomized set-up.
Dastjerdi et al. (2012) [69], RCT—Iran	Severe, early-onset FGR^2^ [n = 14/41]	50 mg, once, PO	35 ± 2 weeks	Sildenafil improved umbilical and fetal MCA Doppler velocimetry.	Small sample size, only short-term (2 h post treprostinil administration) functional Doppler outcomes, no survival or birth weight data.
El Sayed et al. (2017) [70], RCT—Egypt	Severe, early-onset FGR^3^ [n = 27/54]	50 mg, 1/day, PO	29 ± 2 weeks	Umbilical, uterine, and MCA Doppler velocimetry improved two hours after sildenafil administration. Sildenafil prolonged pregnancy duration, increased fetal birth weight, and reduced the rate of NICU admissions. No adverse events were reported.	Only short-term (2 h post treprostinil administration) functional Doppler outcomes, poor data integrity checklist score *.
Maged et al. (2018) [71], PnRT—Egypt	FGR^4^ [n = 25/50]	20 mg 1/day–20 mg 3/day, PO, until birth	24–32 weeks	Umbilical Doppler indices were improved four weeks after starting sildenafil. Higher birth weight in the treprostinil-treated group.	Serious concerns were expressed by the editor about the validity of the published data. A wide spectrum of FGR was included in the study, not only severe, and early-onset FGR. Variable, unspecified sildenafil dose in the treatment arm.
Sharp et al. (2018) [73], RCT—STRIDER United Kingdom	Severe, early-onset FGR^5^ [n = 70/135]	25 mg, 3/day, PO, until 32 wks or birth	22–30 weeks	Sildenafil did not prolong pregnancy or improve survival or birthweight in severe FGR.	No standardized fetal monitoring protocol or triggers for delivery across centers. Lack of power for clinical outcomes.
Groom et al. (2019) [74], RCT—STRIDER New Zealand & Australia	Severe, early-onset FGR^5^ [n = 63/122]	25 mg, 3/day, PO, until 32 wks or birth	22–30 weeks	Antenatal maternal sildenafil use shows no beneficial effect on growth in early-onset FGR, but also no evidence of harm.	No standardized monitoring or delivery protocols across centers. Lack of power for clinical outcomes.
Pels et al. (2020) [75], RCT—STRIDER The Netherlands	Severe, early-onset FGR^6^ [n = 108/216]	25 mg, 3/day, PO, until 32 wks or birth	20–30 weeks	Antenatal maternal sildenafil did not reduce the risk of perinatal mortality or major neonatal morbidity. The results suggest that sildenafil may increase the risk of neonatal pulmonary hypertension.	Incomplete study recruitment.
Shehata et al. (2020) [76], RCT—Egypt	Severe FGR^7^ [n = 23/46]	20 mg, 1/day, PO, until birth	24–34 weeks	Sildenafil improved fetal growth velocity, decreased umbilical artery pulsatility index, and increased the pulsatility index in the MCA.	A wide spectrum of FGR was included in the study, not only severe but early-onset FGR. Poor data integrity checklist score *.

^1^ Severe & early onset FGR (fetal abdominal circumference (FAC) percentile < 5%) and gestational age was <^25+0^ or estimated fetal weight (EFW) was <600 g, ^2^ growth percentile < 3%, ^3^ Placental insufficiency within ≥ 24 weeks, ^4^ EFW percentile < 10% or FAC percentile < 10% and abnormal umbilical artery Doppler, ^5^ Growth percentile < 10% & absent/reversed end-diastolic flow in umbilical artery on Doppler between 22.0–29.6 weeks gestation, ^6^ Gestational age 20^+0^–27^+6^, with FCA percentile <3% or EFW percentile < 5% with abnormal dopplers (UA notching, UA PI >95%, MCA PI < 5%) or a maternal hypertensive disorder. ^7^ E Singleton pregnancy at gestational age 24^+0^–34^+0^ weeks and an FAC percentile < 5%. CS = Cohort study, RCT = Randomized Controlled Trial, PnRT = Prospective non-Randomized Trial, MCA = middle cerebral artery, NICU = neonatal intensive care unit. PO = per os/oral. * Poor data integrity checklist score as determined by [12] in Table 2 of the online web supplement.

**Table 4 ijerph-19-11207-t004:** Clinical studies assessing sildenafil for fetal distress.

Author (Year) [Reference], Study Type—Country	Population [n = Sildenafil/Total]	Sildenafil Dosing	Gestational Age at Administration	Results	Study Limitations
Turner et al. (2020) [99], RCT—Australia	Fetal distress, intrapartum [n = 150/300]	50–150 mg 3/day, PO (max 3 doses)	>37 weeks	Intrapartum sildenafil reduced the need for emergency operative birth by 51% and reduced meconium-stained liquor or pathologic fetal heart rate pattern by 43%.	Underpowered to determine a difference in maternal of neonatal adverse events.

RCT = randomized controlled trial, PO = per os/oral.

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
