# Peer review of "Sildenafil during the 2nd and 3rd Trimester of Pregnancy: Trials and Tribulations"

_ijerph, 2022, doi:10.3390/ijerph191811207_

Round 1
Reviewer 1 Report
This is a really interesting and well-written article. I have a few minor suggested changes.
Few un-reference sentence throughout the text that should ideally have citations.
Consider specifying in the title that your review is of second and third trimester exposure.
“demonstrated prolongation of pregnancy by four days” – should this be an average of 4 days? A confidence interval or similar statistic might also be helpful here.
Did the study by Trapani et al (2016) (and several others) really have no limitations?
Despite limitation on the abstract word count, I think it would be appropriate to include the conditions being examined to assist with searching for your article. For example, someone looking for papers on treatments for FGR may not find your paper as FGR is not part of the abstract/title/keywords etc.
Author Response
Response to Reviewer 1 Comments
This is a really interesting and well-written article. I have a few minor suggested changes.
Point 1: Few un-reference sentence throughout the text that should ideally have citations.
Response 1: We thank the reviewer for this comment. We have read the manuscript again carefully and have added references to the statements made in lines 131, 186, 217 and 353.
Point 2: Consider specifying in the title that your review is of second and third trimester exposure.
Response 2: We appreciate the reviewer’s comment and agree that the title could be more specific written as: “Sildenafil during the 2nd and 3rd trimester of pregnancy: trials and tribulations.” We have adapted the title accordingly.
Point 3: “demonstrated prolongation of pregnancy by four days” – should this be an average of 4 days? A confidence interval or similar statistic might also be helpful here.
Response 3: We thank the reviewer for this careful observation and have corrected the phrasing as suggested: “demonstrated prolongation of pregnancy by on average four days”.
Point 4: Did the study by Trapani et al (2016) (and several others) really have no limitations?
Response 4: We thank the reviewer for bringing this to our attention. We have formulated study limitations for trapani et al (2016), sharp (2018) and groom (2019) in the corresponding tables.
Point 5: Despite limitation on the abstract word count, I think it would be appropriate to include the conditions being examined to assist with searching for your article. For example, someone looking for papers on treatments for FGR may not find your paper as FGR is not part of the abstract/title/keywords etc.
Response 5: We thank the reviewer for this suggestion. We have added the indications to the abstract, while still remaining under the 200-word limit. We have also added the indications to the keywords to further improve the ‘findability’ of our review.

Reviewer 2 Report
The authors review the using sildenafil during the second and third trimesters of pregnancy, relevant experimental and clinical studies. It’s a topic of interest but the manuscript needs significant improvement before publication. My detailed comments are below:
1. Authors should identify the commonalities of the diseases applied and summarize the characteristics of the pathophysiology.
2. The authors should separately summarize the side effects of sildenafil and the possible complications and risks after its application.
3. The effects on the fetus should be highlighted.
4. The authors should similarly list therapeutic agents similar to sildenafil and explain the advantages and characteristics of sildenafil.
5. Limitations of the study need to be added.
6. For a summary of the literature on whether there are adverse effects in newborns after sildenafil application, please review
Author Response
Response to Reviewer 2 Comments
The authors review the using sildenafil during the second and third trimesters of pregnancy, relevant experimental and clinical studies. It’s a topic of interest but the manuscript needs significant improvement before publication. My detailed comments are below:
Point 1: Authors should identify the commonalities of the diseases applied and summarize the characteristics of the pathophysiology.
Response 1: We thank the reviewer for this comment. In addition to the pathophysiology detailed per indication, we have added an introductory statement in lines 96 – 109 regrouping the different introductions according to disease commonalities:
“While sildenafil has been increasingly proposed and investigated for the treatment of female infertility [15-17] and the prevention of recurrent miscarriages [18-21], this review only discusses the use of sildenafil in the second and third trimester of pregnancy. For most indications, sildenafil was considered because of its local vasodilatory effect to address compromised pulmonary (maternal pulmonary hypertension) or uteroplacental perfusion (preeclampsia, fetal growth restriction, and fetal distress). Its antiremodeling effect was hypothesized to reduce vascular pulmonary (congenital diaphragmatic hernia) or uteroplacental (preeclampsia) maldevelopment. Because of its smooth muscle relaxant effect, sildenafil was also tried as tocolytic for preterm labor.
Hereunder we review the pathophysiologic rationale for the use of sildenafil per indication, as well as the most relevant experimental and clinical studies.”
Point 2: The authors should separately summarize the side effects of sildenafil and the possible complications and risks after its application.
Response 2: We thank the reviewer for this comment.
We have summarized the side effects of sildenafil reported in the literature on line 66-68:
“dose-congruent side-effects including headaches, flushing, muscle cramps, indigestion, transient disturbances of color vision perception (due to weak inhibition of PDE6 in retinal photoreceptors) and penile erections [7].”.
Side-effects of sildenafil relevant to pregnancy were specifically discussed in the ‘2.10 safety and tolerance of sildenafil during pregnancy’-section, which we have extended in response to the reviewer’s comment, lines 385-401:
“Whether sildenafil during pregnancy is safe was addressed in two clinical systematic reviews. Both studies agreed on the occurrence of mild maternal side effects i.e. headaches, flushing, visual disturbances, dizziness, palpitations, arthralgia, dyspepsia and epigastric pain, however not an increased risk compared to placebo [12,122]. Nasal bleeding was the only side-effect that was significantly more associated with sildenafil therapy (RR 10.53, 95% CI 1.36-81.3) [12]. Hypotension, a maternal side-effect of sildenafil that was anticipated, was not reported [122]. Obstetric outcomes depended on the underlying pathologic condition. Sildenafil reduced the risk of operative birth when used for intrapartum fetal distress and was not associated with new-onset pre-eclampsia, FGR or antepartum hemorrhage [12]. Fetal and neonatal outcomes of prenatal sildenafil also were dependent on the underlying condition as detailed above. Prenatal sildenafil was associated with increased risk of persistent pulmonary hypertension only for FGR, because of the findings in the Dutch STRIDER arm [12,82]. Meta-analysis could not find evidence for increased perinatal or neonatal mortality and morbidity (respiratory distress syndrome, necrotizing enterocolitis, intracranial hemorrhage, and retinopathy of the premature) associated with the use of PDE5-inhibitors during pregnancy [12].”
Point 3: The effects on the fetus should be highlighted.
Response 3: We thank the reviewer for this comment and have added a paragraph to discuss the findings of the systematic review by Turner et al, on the fetal effects of sildenafil, lines 395-401:
“Fetal and neonatal outcomes of prenatal sildenafil also were dependent on the underlying condition as detailed above. Prenatal sildenafil was associated with increased risk of persistent pulmonary hypertension only for FGR, because of the findings in the Dutch STRIDER arm [12,82]. Meta-analysis could not find evidence for increased perinatal or neonatal mortality and morbidity (respiratory distress syndrome, necrotizing enterocolitis, intracranial hemorrhage, and retinopathy of the premature) associated with the use of PDE5-inhibitors during pregnancy [12].”
Point 4: The authors should similarly list therapeutic agents similar to sildenafil and explain the advantages and characteristics of sildenafil.
Response 4: Although the review strictly covers prenatal sildenafil use, we agree it would be useful to mention other PDE5-inhibitors and have therefore added the following paragraph line 84-94:
“Sildenafil only modestly reduces blood pressure, granting it a particularly desirable profile for pregnancy where systemic hypotension is deleterious for the fetus [7,9]. Moreover, the hepatic CYP3A4 and CYP2C19 isoenzymes involved in sildenafil’s metabolization were demonstrated to be present and active prenatally [10,11]. Tadalafil, vardenafil, mirodenafil, avanafil and udenafil are other PDE5 inhibitors, however only tadalafil has also been trialed during pregnancy for fetal growth restriction [12,13]. Compared to sildenafil, clinical experience with tadalafil is very limited and immaturity of the glucuronidation pathway used for its metabolization forms a relative contraindication for its use during pregnancy [14].”
Point 5: Limitations of the study need to be added.
Response 5: We have added a paragraph to the main text discussing the limitations of the review on lines 403-415, stating:
2.11 Limitations
This review is limited in that it is not a systematic review in which the Preferred Reporting Items for Systematic Reviews and Meta‐Analyses (PRISMA) guidelines were not strictly adhered to, only the Medline database using PubMed was queried and we also discussed non-randomized clinical trials.
Point 6: For a summary of the literature on whether there are adverse effects in newborns after sildenafil application, please review
Response 6: We thank the reviewer for this suggestion and agree the effects on newborns are of utmost interest to this review. We have therefore expanded the paragraph on the discussion of fetal and neonatal effects of prenatal sildenafil on lines 395-401:
“Fetal and neonatal outcomes of prenatal sildenafil also were dependent on the underlying condition as detailed above. Prenatal sildenafil was associated with increased risk of persistent pulmonary hypertension only for FGR, because of the findings in the Dutch STRIDER arm [12,82]. Meta-analysis could not find evidence for increased perinatal or neonatal mortality and morbidity (respiratory distress syndrome, necrotizing enterocolitis, intracranial hemorrhage, and retinopathy of the premature) associated with the use of PDE5-inhibitors during pregnancy [12].”

Round 2
Reviewer 2 Report
The authors addressed my concerns and my recommendation is accept.
Author Response
Thank you.